# Development and validation of the health education demand scale for HPV infected patients based on KANO model

Yuying Zhang[1,2], Li Luo[1,2], Shuaihui Du[1,2☯], Die Hou[1,2☯], Yaling Zhang[1,2☯], Limin He[1,2☯], Lihua Zhou[1,2]*

**1** Department of VIP Clinic Nursing, West China Second University Hospital, Sichuan University/West China School of Nursing, Sichuan University, Chengdu, Sichuan, China, **2** Key Laboratory of Birth Defects and Related Diseases of Women and Children (Sichuan University), Ministry of Education, Chengdu, Sichuan, China

☯ These authors contributed equally to this work.

* 1424170780@qq.com

**Data Availability Statement:** All relevant data are within the paper and its Supporting Information files.

## Abstract

### Objective

The purpose of this study is to develop and validate the scale of health education demand of patients with HPV infection based on KANO model, so as to provide a tool for further exploring the types of health education demand and influencing factors of patients with HPV infection.

### Methods

This study is a scale development and validation study using a three-stage cross-sectional design. In stage 1, a preliminary item pool is formed using literature review, semi-structured interviews and the Delphi method. In stage 2, six experts were invited to assess content validity. A cross-sectional survey was conducted on 1169 patients with HPV infection, Questionnaire results from 583 patients were used for exploratory factor analysis. In stage 3, the remaining 586 patients to validate the factor structure through confirmatory factor analysis.

### Results

In stage 1, an initial 35-item scale was developed and the items were transformed positive and reverse based on KANO model. In stage 2, Exploratory factor analysis formed a scale of 28 items in 5 factors: disease information demand, social support, emotional demand, family support and health education style demand. Cronbach's alpha was 0.940 for the entire scale and 0.763~0.908 for the five subscales in the positive items, 0.955 for the entire scale and 0.739~0.946 for the five subscales in reverse items. The content validity index of the scale: S-CVI/UA = 0.91, S-CVI/Ave = 0.98. In stage 3, the confirmatory factor analysis showed that the $\chi2/df$, RMSEA, CFI and TLI of the positive items after four model modifications were 3.650, 0.067, 0.901, 0.888, and the SRMR value was < 0.001. The fitting of the five-factor model was good.

**Funding:** The author(s) received no specific funding for this work.

**Competing interests:** The authors have declared that no competing interests exist.

## Conclusion

The KANO model based questionnaire on health education demand of HPV infected patients has good reliability and validity, and is suitable for the investigation of health education demand of HPV infected patients.

## Introduction

Human Papillomavirus (HPV) is a virus with multiple genotypes, which is closely linked to the human reproductive system lesions and the development of malignant tumors. Among them, continuous infection of high-risk HPV is a major cause of cervical cancer in women [1]. In China, the female HPV infection rate was 16.18%, and the high-risk infection rate was 12.95% [2]. Among the high-risk HPV (HR-HPV) infection subtypes, HPV16 and HPV18 had the highest infection rate in different regions and different populations [3].

HR-HPV infection is closely related to the occurrence of cervical precancerous lesions. The results of a meta-analysis showed that the infection rate of HR-HPV increases with the serious cervical precancerous lesions. In normal women, HPV16, 58, 52, 18, 39 and 33 with infection rates ranging from 0.9% to 3.27%. In ICC patients, the infection rate of HPV16 was as high as 58.7%, followed by HPV18, 58, 33, 52 and 31, with the infection rate ranging from 2.3% to 11.0% [4]. This is similar to the findings of Jing He [5]. The higher infection rate of HPV, More severe precancerous lesions are associated with higher rates of HPV infection.

Additionally, the age-specific distribution of HPV infection showed a bimodal curve, and the infected patients are becoming younger and younger [6]. The HPV infection rate of young women under 25 years old reached 36.22% [7]. At the same time, cervical cancer patients are also gradually getting younger, study found that the average age of onset of cervical cancer increased by 4.8 years in six years [8], and the incidence of cervical cancer in women under 35 years old has increased significantly. And although the age of onset of cervical cancer is mainly concentrated in 35–59 years old, the youth group (15–34 years old) is still on the rise [9]. Moreover, the mortality rate of cervical cancer in women between 30 and 59 years old in eastern China is also on the rise, which is closely linked to high-risk HPV infection [10].

The increasing incidence and the tendency of younger age make it more urgent for the public to increase the demand for HPV-related information. There are nearly 10,000 records about HPV vaccine through online social networking and question-and-answer community platforms [11]. Compared with 10 years ago, the awareness of cervical cancer, HPV and HPV vaccine in China has improved, but the awareness is still limited, and the good awareness rate of HR-HPV-infected women is only 42.5% [12]. Most of the women self-perceived low awareness of cervical cancer risk factors and their prevention, but they expressed a positive attitude toward HPV screening and asked for more information about cervical cancer and its prevention [13].

To clarify the demand of patients is to provide them with health assessment, analysis, health advice and guidance from different paths, and the initial steps of treatment and intervention for related risk factors, which helps patients better participate in the formulation and implementation of health management plans [14]. Therefore, how to accurately and effectively assess the health management demand of patients with HPV infection is particularly important. The age distribution of patients is extensive, and people of different ages have different understandings of diseases and needs for health education. Patients often face both physical

and psychological pressure, a specific health education information demand scale is urgently needed to assess the health education demand of different patients.

KANO model is a simple and easy technology to identify the attributes of service demand, which divides the attributes of products into five types according to the relationship between the objective performance of products and the subjective feelings of customers: Attractive Quality, Must-be quality, One-dimensional Quality, Indifferent quality and Reverse quality are used to formulate quality management strategies and improvement plans [15]. In the field of health education services, they have been widely applied to pregnant women, tumor patients and other groups.

Therefore, the purpose of this study is to develop a scale of health education service demand for patients with HPV infection, and to provide theoretical basis for the implementation of personalized and precise health education programs in the future through KANO model.

## Methods

### Study design

A three-stage cross-sectional design was adopted in this study. In stage 1, the first draft of the scale was developed through a literature review, semi-structured interviews, and the Delphi method. In stage 2 and stage 3, cross-sectional survey was conducted to measure and validate the reliability and validity of the health education demand scale for HPV infected patients based on KANO model. We followed GRRAS checklist to report this study.

### Stage 1: Scale development

**Item generation.** Based on the social ecosystem theory, which has been widely used in the construction of assessment tools for human-related health factors, including chronic, psychological problems, health promotion policy formulation and other fields [16–18].

We defined the health education demand of the patients with HPV, which emphasize the interaction between systems and individuals and the influence of each system on human development [19]. Such as the demand in physical, psychological and social aspects.

In the literature review, we systematically searched CNKI, Wanfang, PubMed, embase and other databases, read and collected literatures related to the demand of patients with HPV infection for literature review, and obtained relevant topics related to the health education demand of patients with HPV infection in combination with social ecosystem theory. Subsequently, the interview outline of qualitative interview was developed through the method of expert meeting, and the inclusion and exclusion criteria were established. Inclusion criteria for the patients who (1) aged ≥16 years; (2) met the relevant diagnostic criteria for cervical lesions in Gynecology and Obstetrics [20], and were confirmed as cervical lesions by the pathological examination of Thinprepcy tologictest (TCT). (3) could communicate normally and cooperate with the investigation and evaluation; (4) Informed, voluntary participation in this study, signed informed consent. Exclusion criteria for the patients who (1) with other malignant tumors; (2) with AIDS, syphilis and other infectious sexually transmitted diseases and with severe organ function injury; (3) with fungal vaginitis, bacterial vaginitis and other cervical inflammatory diseases.

**Sample/participants and data collection.** From September 19, 2023 to October 19, 2023, HPV infected patients were selected for interview, and the sample size was based on the principle of data saturation. All patients signed written informed consent forms and completed the interviews. At the same time, the consent of the parents or guardians of the minors involved in the study was obtained. Finally, 11 interviewees were included. The interview outline is as follows. (1) What are your current needs in health education? (2) What kind of health education

support do you expect medical staff to provide in the process of seeing a doctor? (3) What kind of health education guidance do you want after the visit? (4) What problems do you think still exist in health education? (5) What health education related services do you think will affect your satisfaction? The rooted theory was adopted for coding, and the needs characteristics of core categories and subcategories were extracted according to the needs mentioned by patients in the interview. On the basis of literature analysis and semi-structured interview, a 45-item questionnaire on health education demand of patients with HPV infection was formed by referring to relevant demand scale indicators and combining with the background of clinical nursing work in China through brainstorming and modification by the research group.

**Expert letter.** From October 20, 2023 to November 20, 2023, According to the content of this study, 15 experts were selected from related fields of gynecology, nursing and psychology. Based on the principles of professional complementarity, reasonable knowledge structure, research purpose, expert authority and representativeness, 15 experts were invited for correspondence consultation. Inclusion criteria: (1) Bachelor degree or above; (2) Intermediate and above professional titles; (3) More than 10 years of relevant work experience; (4) Have a high academic level in the field of obstetrics and gynecology clinical medicine or obstetrics and gynecology nursing, especially in the diagnosis and treatment of cervical diseases; (5) Familiar with the preparation process of the scale. The experts signed written informed consent and agreed to participate voluntarily.

The expert correspondence questionnaire consists of three parts. (1) Questionnaire description: including background, theoretical concept, research purpose. (2) Questionnaire on health education demand of patients with HPV infection: The importance of each item was rated on a 5-point Likert scale (1 = not important and 5 = very important), and the columns of "Modified opinions", "deleted opinions" and "Proposed to add indicators" were set. (3) The basic information table of the experts: including the name, age, educational background, professional title, and the expert's familiarity with this research question and the basis for judgment.

In this study, questionnaires were sent out by email and asked for responses within 2 weeks. After the first round of consultation, the research group made modifications according to the results of the consultation, formed the second round of consultation questionnaire, and then sent it to the experts. After the second round of consultation, if the expert opinions converge, the consultation will be ended, otherwise it will continue. The index screening criteria of the letter inquiry results were as follows: the indexes with importance assignment ≤3.5 points, coefficient of variation > 0.3 and full score <20% were excluded.

**KANO model questionnaire preparation.** Through previous literature review, qualitative interview and expert letter consultation, the scale items of health education demand of patients with HPV infection was developed, and then Kano transformation practice was carried out. Based on the finally established questionnaire item pool, positive items and reverse items on health education demand of patients with HPV infection were set according to the characteristics of KANO model questionnaire. This kind of two-way question means that for each requirement item, you set the "provide, how do you feel?" With "don't provide, how do you feel?" The alternative answers were "like (5 points)", "take for granted (4 points)", "don't care (3 points)", "can tolerate (2 points)", "don't like (1 point)".

**Face validity.** According to the inclusion and exclusion criteria, From November 21, 2023 to November 30, 2023, 20 patients with HPV infection were selected to evaluate the difficulty, suitability and ambiguity of each item. They signed written informed consent and agreed to participate voluntarily. During the investigation, the difficult problems in the items were modified and adjusted, and the draft of the health education demand scale for patients with HPV infection was developed through language debugging to ensure the comprehensibility of the scale.

## Stage 2: The first cross-sectional survey

**Sample and setting.** From December 1, 2023 to March 31, 2024, a questionnaire survey was conducted on patients with HPV infection in a Grade A obstetrics and gynecology hospital in Chengdu, China. According to the Kendall criterion, item analysis requires the sample size to be 5 ~ 10 times the number of items in the scale. Factor analysis requires a sample size of 10 to 20 times the number of items on the scale, and the follow-up loss of 10% was considered [21, 22]. The number of items in the scale is 35. The sample size of n1 = 350–700 was calculated. In the end, n1 = 583 women were included because the data from exploratory factor analysis (EFA) could not be duplicated with those from confirmatory factor analysis (CFA), and n2 = 586 patients were included considering that the sample size should not be less than 200 and in accordance with the principle that CFA should be greater than EFA [23, 24]. Therefore, a total of 1169 HPV-infected patients were recruited to participate in the scale survey. The inclusion and exclusion criteria are the same as for semi-structured interviews.

**Instruments.** The research tool used a general information questionnaire designed by the research team to collect basic information about the respondents, including demographic information (age, occupation, educational level, marital status and economic status, etc.) and disease-related information (disease diagnosis, sexual frequency, number of sexual partners, etc.).

**Data collection methods and quality control.** The patients to be included in the study were identified through the hospital big data system, and the hospital follow-up platform was contacted to conduct the survey by sending questionnaires through the hospital public account and short message. Two managers of the hospital follow-up platform were given unified training to modify the content to be adjusted in the online questionnaire, and the time for sending and retrieving questionnaires was set. The data were exported by the managers of the follow-up platform every month. For those who did not complete all the questions, the researchers contacted the patients by phone to ask about their results, and for those who did not complete the questionnaire, the system would resend it every other month. After the data were exported, two staff members screened out the questionnaires that took less than 5 minutes to fill in and had consistent answers or obvious rules and inconsistent answers. All patients signed informed consent forms and completed questionnaires on the follow-up platform. The consent of the parents or guardians of the minors involved in the study was also obtained.

**Data analysis.** The data was analyzed using IBM SPSS Statistical window (version 20.0), and IBM SPSS AMOS (version 23.0). The first survey data (n1 = 583) were used for item analysis, reliability test, content validity test and exploratory factor analysis. Qualitative data were represented by frequency and component ratio (%), while quantitative data conforming to normal distribution were represented by mean ± standard deviation. P<0.05 indicated that the difference was statistically significant.

**Project analysis.** For item analysis, (1) critical ratio method was used to calculate the scores of each item of the scale, rank the scores from high to low, compare the scores between the high group (top 27%) and the low group (bottom 27%), and delete the items with a decision value < 3 and no statistically significant difference (P<0.05) [25]. (2) Correlation coefficient method: The correlation between each item and the total score of the scale was calculated, and the items with no statistical significance (P > 0.05) or < 0.3 were deleted [26]. (3) Cronbach's alpha coefficient method: After deleting an item, if the Cronbach's alpha coefficient of the scale is significantly increased, the item will be deleted [25]. The coefficients of positive and reverse items of health education demand of patients with HPV infection were calculated respectively.

**Reliability test.** The reliability test adopts internal consistency reliability and retest reliability. The internal consistency of the scale was measured using the Cronbach's Alpha coefficient method, and retest reliability was assessed by calculating the coefficient of correlation between the scale data collected 4 weeks after the start of the survey and the data from the first measurement. Generally, Cronbach's alpha coefficient and broken half reliability are required to be > 0.7 [27]. Retest reliability is generally required to be above 0.7. The coefficients of positive and reverse items of health education needs of patients with HPV infection were calculated respectively.

**Validity test.** *Structural validity*. KMO and Bartlett tests were used to verify the validity of positive and reverse items in the KANO attribute questionnaire, respectively. Under the premise that KMO value was >0.5 and the Bartlett's sphericity test showed P<0.05, exploratory factor analysis was carried out, and principal component analysis and variance maximization orthogonal rotation method were used. The statistical criteria for factor extraction and item selection include: (1) the eigenvalue of each factor is greater than 1 and there are at least 2 items; (2) The factor load of each article is greater than 0.5 and at least greater than 0.4; (3) An entry cannot have a high load on two or more factors at the same time (both above 0.4 and the difference is less than 0.2) [25, 28];

*Content validity*. The importance of each item of the scale and its correlation with the corresponding contents were evaluated according to the experts, and the content validity index was calculated. 6 experts with good authority in expert correspondence were selected to evaluate whether the scale included enough and appropriate items, and whether the scale items were related to the topic The options were 'not relevant = 1', 'somewhat relevant = 2', 'relevant = 3', and 'very relevant = 4'. Content validity index (I-CVI) at the item level and content validity index (S-CVI) at the scale level were used for evaluation, and S-CVI was divided into consensus S-CVI (S-CVI/UA) and average S-CVI (S-CVI/Ave), the scale I-CVI is not less than 0.78, and the S-CVI/UA and S-CVI/Ave are not less than 0.8 and 0.9, respectively, indicating good content validity [29].

## Stage 3: The second cross-sectional survey

**Sample and setting.** Based on patient recruitment in stage 2. We used n2 = 586 patients for confirmatory factor analysis (CFA), which exceeded the suggested minimum sample size of 200 for confirmatory factor analysis (CFA) [30].

**Instruments.** We collected the same demographic and clinical information, including demographic information (age, occupation, educational level, marital status and economic status, etc.) and disease-related information (disease diagnosis, sexual frequency, number of sexual partners, etc.).

The second version of the scale was developed after the EFA in Stage 2, which contained five dimensions and 28 items in total. The first dimension was disease information demand (nine items), social support (four items), emotional demand (three items), family support (three items) and health education style demand (nine items).

**Data collection.** This data were collected between February 1, 2024 and March 31, 2024. All eligible patients were invited to fill in a questionnaire. All patients signed informed consent forms and completed questionnaires on the follow-up platform. The consent of the parents or guardians of the minors involved in the study was obtained. The questionnaire is anonymous and no personal information is collected. These questions are double-checked at completion to minimize lost data.

**Confirmatory factor analysis.** Sample 2 was used for confirmatory factor analysis. Confirmatory fitting criteria based on Kline [31]: standardized chi-square statistics ($\chi 2$ / df; <5.0),

comparative fit index (CFI; ≥0.9), Tuck-Lewis index (TLI; ≥0.9), approximate root mean square error (RMSEA; ≤0.08) and standardized root mean square residuals (SRMSR; ≤0.08).

**Ethical considerations.** The study was carried out in accordance with the principles of the Declaration of Helsinki. This study was approved by the Medical Ethics Committee of West China Second Hospital of Sichuan University (No. 230 of the 2023 round of medical research approval).

## Results

### Results in stage 1

**Delphi expert correspondence results.** A total of 15 experts were included in this study, including clinical medical experts and nursing experts. 15 questionnaires were sent out in two rounds of expert correspondence, and 15 were recovered, with a questionnaire recovery rate of 100%. Among them, 53.3% of the experts held a master's degree or above, 60% were nurses. The research field includes cervical diseases, clinical nursing of cervical diseases, nursing education, nursing management, nursing research, and psychological nursing. The coefficient of expert familiarity in the two rounds of correspondence was 0.853, the coefficient of judging basis was 0.907 and 0.910, and the coefficient of authority was 0.880 and 0.882. Kendall harmony coefficient was 0.239 and 0.240. The correspondence results were consistent.

**Item extraction.** Through literature review and semi-structured interviews, we extracted 45 items.

According to the first round of expert letter consultation, it is considered that items "provides health education on the mode of HPV transmission", "inform the precautions for preventing HPV infection in life (sexual behavior, diet, exercise, environment, etc.)", "inform HPV infection or relevant guidance on sexual life after treatment", "inform patients of the impact of the disease on family members," and "Inform patients of the impact of the disease on family members to avoid cross-infection" are repeated, so the four items are combined as "Provide health guidance on HPV prevention, transmission methods, sexual life, etc.". In addition, items "provide health education on pathogenesis and treatment (treatment time, cycle, treatment plan)", "provide drug guidance related to HPV infection", "inform the effect of HPV infection after treatment and the condition of negative conversion", "provide privacy protection", "provide a comfortable environment for medical treatment" and "provide sanitary napkins, tampons, condoms, wheelchairs, reading glasses, bread, snacks" are not suitable to be deleted as the content of health education needs. Replace item "Inform patients of the doctor's disease areas of expertise" with "Provide access to the doctor's disease areas of expertise".

The second round of expert consultation was again considered and refined on the basis of the first round, and it was considered that the item "provide health guidance on HPV prevention, transmission methods, sexual life, etc." was three different needs and should be elaborated separately. Experts suggested to add an item of "provide relevant information about the treatment of myself and my sexual partner". The sentences "provide regular follow-up service (telephone consultation)" was refined to "provide regular health education follow-up service"; The items" online public accounts push HPV infection science education", "open online psychological communication and consultation", were revised and changed to "online HPV infection science education articles" and "open online psychological consultation and communication" respectively.

After two rounds of expert letter consultation, a questionnaire for health education demand of HPV infected patients with a total of 35 items was finally formed. The preliminary investigation results showed that the scale was simple to operate, and the patients needed about 8-15min to complete the scale.

### Results in stage 2

**Participants' characteristics and descriptive statistics.**   The total sample size of this study was n = 1169, among which n1 = 583 and n2 = 586 were collected in different periods and were not the same samples. The average age of HPV-infected patients was 41.78 years. Most people living in cities (77.2%), most patients married (75.2%), more than 80% of married people are cohabiting, and the frequency of sexual life 1–3 times per month accounted for the majority (77.5%), more than 50% of unmarried people have sexual partners, the frequency of sexual life 1–3 times per month accounted for the majority (63.6%). 85% of the patients had their first sexual experience between the ages of 16 and 25, 34.1% had more than three sexual partners, and 49.5% had more than three pregnancies. In addition, 44.4% of the patients had low frequency of HPV screening (< 1 time/year), and 18.5% of them had never had HPV screening, more details were shown in Table 1.

**Project analysis result.**   The results of the critical ratio method showed that the difference between the items in the high and low groups of positive items was statistically significant (P < 0.001), and the T-values ranged from 6.226 to 20.613, all of which were >3.000. The correlation coefficients between the items and the total scale ranged from 0.482 to 0.730, and the correlation coefficients were all > 0.400, with significant correlation coefficients (P < 0.001). The homogeneity between the items and the whole scale was high. The difference between the high and low groups of the reverse items was statistically significant (P < 0.001), and the T-value ranged from 8.660 to 41.428, all of which were >3.000, indicating good differentiation of the items. The correlation coefficients between the items and the total scale ranged from 0.467 to 0.786, and the correlation coefficients were all > 0.400, with significant correlation coefficients (P < 0.001). The homogeneity between the items and the whole scale was high.

**Validity analysis results.**   The results of exploratory factor analysis showed that the KMO value of the positive item was 0.950 and the Bartlett's sphericity test $\chi^2$ value was 12013.8 (P<0.001); the KMO value of the reverse item was 0.965 and the Bartlett's sphericity test $\chi^2$ value was 15703.79 (P<0.001). It is suitable for factor analysis. By using principal component analysis and maximum variance orthogonal rotation method, the results showed that 5 common factor eigenvalues were > 1 in the positive items, and the cumulative variance contribution rate was 59.050%; 5 common factor eigenvalues were > 1 in the reverse items, and the cumulative variance contribution rate was 67.191%; in the positive items, the load values of items 11, 20, 23 and 26 were less than 0.5. After deleting item 11, item 5 with load value < 0.5 will also be deleted; Items 13 and 17 have double loads and the load difference is less than 0.15, so they are deleted. After the items are deleted according to the load value of the positive items, the eigenvalues of the reverse items become 4 common factors > 1. The eigenroots of the positive and reverse factors and the contribution rates of variance are shown in Table 2. The results of exploratory factor analysis for positive and reverse items are shown in Tables 3 and 4. Table 3 shows the load values of the positive items, the corresponding load values of each item is > 0.5, and the items are divided into 5 dimensions according to the load values. Each dimension is named as disease information demand (9 items), social support (4 items), emotional demand (3 items), family support (3 items) and health education style demand (9 items). Table 4 shows the load values of the reverse items, and the corresponding load values of each item is > 0.4, the items 8, 10 and 14, which are classified as the second dimension according to the load values. Item 19 is attributed to the third dimension, and the rest fit the factor load attribution dimension of the positive items.

**Table 1. Sample characteristics (N = 1169).**

| Characteristics | | n | % |
|---|---|---|---|
| Age (range: 16-76years) | Mean = 41.78years (SD = 10.27) | | |
| Residence | village | 179 | 15.4% |
| | suburb | 77 | 6.6% |
| | City | 900 | 77.2% |
| | Other | 10 | 0.9% |
| Cervical cancer screening frequency | Never | 216 | 18.5% |
| | Once every 3 years or more | 154 | 13.2% |
| | Every two years | 148 | 12.7% |
| | Once a year | 499 | 42.8% |
| | Multiple times a year | 148 | 12.7% |
| Marital status | Unmarried | 110 | 9.5% |
| | Divorced | 150 | 12.9% |
| | Widow | 29 | 2.5% |
| | Married | 875 | 75.2% |
| lifestyle | cohabitation | 704 | 80.8% |
| | Live apart | 146 | 16.8% |
| | Other | 21 | 2.4% |
| Sexual frequency | 1–3 times a month | 606 | 77.5% |
| | 1–3 times a week | 147 | 18.8% |
| | 4–6 times a week | 21 | 2.7% |
| | 7 or more times a week | 8 | 1.0% |
| Whether have a sexual partner | Yes | 168 | 59.2% |
| | No | 116 | 40.8% |
| Frequency of sexual intercourse with sexual partners | 1–3 times a month | 103 | 63.6% |
| | 1–3 times a week | 45 | 27.8% |
| | 4–6 times a week | 13 | 8.0% |
| | 7 or more times a week | 1 | 0.6% |
| Age of first sexual intercourse | never | 3 | 0.3% |
| | <16 | 35 | 3.0% |
| | 16~25 | 989 | 85.0% |
| | 26~35 | 127 | 10.9% |
| | >35 | 9 | 0.8% |
| Frequency of oral contraceptives | Less than once a month | 197 | 91.6% |
| | 1–3 times a month | 9 | 4.2% |
| | 3 or more times per month | 9 | 4.2% |
| Number of pregnancies | Never | 132 | 11.4% |
| | 1 | 191 | 16.5% |
| | 2 | 260 | 22.5% |
| | ≥3 | 572 | 49.5% |
| Number of sexual partners | 1 | 518 | 44.9% |
| | 2 | 242 | 21.0% |
| | ≥3 | 393 | 34.1% |

After consultation by six experts, the content validity index of each item was 0.833~1.000, the content validity index of the scale S-CVI/UA = 0.91, S-CVI/Ave = 0.98, and the validity was in line with the standard.

**Table 2. The total variance of the factors (positive and reverse items).**

| factor | positive items | | | reverse items | | |
|---|---|---|---|---|---|---|
| | eigenvalue | variance | cumulative variance | eigenvalue | variance | cumulative variance |
| 1 | 11.514 | 17.985% | 17.985% | 12.966 | 46.308% | 46.308% |
| 2 | 2.282 | 15.574% | 33.559% | 2.892 | 10.329% | 56.636% |
| 3 | 1.330 | 11.129% | 44.688% | 1.373 | 4.902% | 61.538% |
| 4 | 1.171 | 8.907% | 53.595% | 1.088 | 3.886% | 65.424% |
| 5 | 1.010 | 8.220% | 61.814% | | | |

**Reliability analysis results.** The Cronbach's α coefficient of positive items was 0.940, and the Cronbach's α coefficient of each dimension was 0.850, 0.863, 0.783, 0.763, 0.908. The total broken half reliability of the scale was 0.864, and the broken half reliability of each dimension was 0.829, 0.856, 0.785, 0.743 and 0.840. After 4 weeks, the retest reliability of positive entries

**Table 3. Results of exploratory factor analysis of positive items (n1 = 583).**

| Item (28) | Common factor | | | | |
|---|---|---|---|---|---|
| | 1 | 2 | 3 | 4 | 5 |
| 1. How do you feel if you are provided health guidance on HPV prevention? | 0.647 | | | | |
| 2. How do you feel if you are provided with information about HPV vaccine (type, vaccination)? | 0.565 | | | | |
| 3. How do you feel if you are provided about health education on HPV screening (HPV typing, TCT)? | 0.663 | | | | |
| 4. How do you feel if you are provided about providing HPV-related screening and prevention guidance to sexual partners? | 0.542 | | | | |
| 6. How do you feel if you are provided relevant knowledge about the relationship between HPV infection and cervical lesions? | 0.572 | | | | |
| 7. How do you feel if you are provided with guidance for HPV infection related tests (colposcopy, biopsy)? | 0.740 | | | | |
| 8. For HPV, how do you feel if you are told the method and time of the review? | 0.667 | | | | |
| 10. How do you feel if you are provided information about the treatment for yourself and your sexual partner? | 0.548 | | | | |
| 14. How do you feel if you are told about the prognosis (quality of life, recurrence)? | 0.522 | | | | |
| 9. For HPV, how do you feel if you are told about self-care guidance and precautions? | | 0.694 | | | |
| 12. How do you feel if we instruct you to avoid risk factors related to recurrence? | | 0.616 | | | |
| 19. How do you feel if you are informed about the latest developments in the treatment of HPV infection? | | 0.646 | | | |
| 21. How do you feel if you are provided a way to understand doctors' areas of expertise in diseases? | | 0.742 | | | |
| 15. How do you feel if you are provided about health education on fertility possibilities and reproductive decision-making? | | | 0.733 | | |
| How do you feel if you are provided places and platforms for expressing emotional states? | | | 0.783 | | |
| 18. How do you feel if professional psychological counseling and support (individual psychological counseling, group psychotherapy) are provided? | | | 0.627 | | |
| 22. How do you feel if the family members are informed to participate in the management of the disease together? | | | | 0.725 | |
| 24. How do you feel if family members are allowed to accompany them when necessary? | | | | 0.802 | |
| 25. How do you feel if we help patients maintain and build intimate relationships with their partners? | | | | 0.627 | |
| 27. How do you feel if a health education environment with warm environment, suitable temperature and teaching conditions is provided? | | | | | 0.563 |
| 28. How do you feel if you are informed the specific content or time of health education through SMS or push? | | | | | 0.621 |
| How do you feel if HPV-related health education knowledge manuals are distributed? | | | | | 0.657 |
| 30. How do you feel if doctors participate in HPV-related knowledge education? | | | | | 0.607 |
| 31. How do you feel about the case management platform for patients with HPV infection? | | | | | 0.565 |
| 32. How do you feel about online HPV disease-related health consultation? | | | | | 0.674 |
| 33. How do you feel if an article about HPV infection is pushed online? | | | | | 0.787 |
| 34. How do you feel if a short health education video related to HPV infection is pushed online? | | | | | 0.769 |
| 35. How do you feel if we provide online psychological counseling and communication? | | | | | 0.691 |

**Table 4. Results of exploratory factor analysis of reverse items (n1 = 583).**

| Item (28) | Common factor | | | | |
|---|---|---|---|---|---|
| | 1 | 2 | 3 | 4 | 5 |
| 1. How do you feel if you are provided health guidance on HPV prevention? | 0.79 | | | | |
| 2. How do you feel if you are provided with information about HPV vaccine (type, vaccination)? | 0.774 | | | | |
| 3. How do you feel if you are provided about health education on HPV screening (HPV typing, TCT)? | 0.772 | | | | |
| 4. How do you feel if you are provided about providing HPV-related screening and prevention guidance to sexual partners? | 0.705 | | | | |
| 6. How do you feel if you are provided relevant knowledge about the relationship between HPV infection and cervical lesions? | 0.67 | | | | |
| 7. How do you feel if you are provided with guidance for HPV infection related tests (colposcopy, biopsy)? | 0.589 | | | | |
| 8. For HPV, how do you feel if you are told the method and time of the review? | | 0.818 | | | |
| 10. How do you feel if you are provided information about the treatment for yourself and your sexual partner? | | 0.412 | | | |
| 14. How do you feel if you are told about the prognosis (quality of life, recurrence)? | | 0.603 | | | |
| 9. For HPV, how do you feel if you are told about self-care guidance and precautions? | | 0.779 | | | |
| 12. How do you feel if we instruct you to avoid risk factors related to recurrence? | | 0.523 | | | |
| 19. How do you feel if you are informed about the latest developments in the treatment of HPV infection? | | | 0.581 | | |
| 21. How do you feel if you are provided a way to understand doctors' areas of expertise in diseases? | | | 0.499 | | |
| 15. How do you feel if you are provided about health education on fertility possibilities and reproductive decision-making? | | | 0.705 | | |
| 16. How do you feel if you are provided places and platforms for expressing emotional states? | | | 0.692 | | |
| 18. How do you feel if professional psychological counseling and support (individual psychological counseling, group psychotherapy) are provided? | | | 0.646 | | |
| 22. How do you feel if the family members are informed to participate in the management of the disease together? | | | | 0.561 | |
| 24. How do you feel if family members are allowed to accompany them when necessary? | | | | 0.827 | |
| 25. How do you feel if we help patients maintain and build intimate relationships with their partners? | | | | 0.633 | |
| 27. How do you feel if a health education environment with warm environment, suitable temperature and teaching conditions is provided? | | | | | 0.63 |
| 28. How do you feel if you are informed the specific content or time of health education through SMS or push? | | | | | 0.72 |
| 29. How do you feel if HPV-related health education knowledge manuals are distributed? | | | | | 0.757 |
| 30. How do you feel if doctors participate in HPV-related knowledge education? | | | | | 0.75 |
| 31. How do you feel about the case management platform for patients with HPV infection? | | | | | 0.746 |
| 32. How do you feel about online HPV disease-related health consultation? | | | | | 0.792 |
| 33. How do you feel if an article about HPV infection is pushed online? | | | | | 0.83 |
| 34. How do you feel if a short health education video related to HPV infection is pushed online? | | | | | 0.828 |
| 35. How do you feel if we provide online psychological counseling and communication? | | | | | 0.748 |

is 0.984, and the retest reliability of all dimensions is 0.990, 0.580, 0.761, 0.717, 0.624. The Cronbach's α coefficient of the reverse items was 0.955, and the Cronbach's α coefficient of each dimension was 0.895, 0.825, 0.851, 0.739, 0.946. The total broken half reliability of the scale was 0.902, and the broken half reliability of each dimension was 0.815, 0.794, 0.845, 0.777 and 0.907. After 4 weeks, the retest reliability of the reverse entry is 0.697, and the retest reliability of all dimensions is 0.476, 0.444, 0.499, 0.638, 0.673.

The five factors were named as disease information demand, social support, emotional demand, family support and health education style demand according to the positive items. In the exploratory factor analysis of the reverse items, after discussion by experts, Item 8 "For HPV, how do you feel if you are told the method and time of the review?", item 10 "How do you feel if you are provided information about the treatment for yourself and your sexual partner?", item 14 "How do you feel if you are told about the prognosis (quality of life, recurrence)?" are as part of the disease information demand. Item 19 "How do you feel if you are informed about the latest developments in the treatment of HPV infection?" is as part of the social support. Therefore, the scale structure adopts the exploratory factor analysis results of

positive items, and finally forms the total demand table of HPV infection health education in KANO model with five subscales.

### Results in stage 3

**Confirmatory factor analysis results.** The results of confirmatory factor analysis showed that 2/df, RMSEA, CFI and TLI of positive items of HPV-infected patients' health education demand scale were 3.650, 0.067, 0.901 and 0.888, respectively, with SRMR values < 0.001. The 2/df, RMSEA, CFI and TLI of the reverse items are 4.693, 0.079, 0.910 and 0.900 respectively, and the SRMR value is < 0.001. On the whole, the structural model has good fitting degree and good structural validity (S1 Fig).

## Discussion

In recent years, the number of patients with HPV infection has increased year by year, and the number of people who continue to be reinfected after negative transition has also increased. Based on the ecosystem theory [19], this study compiled the scale of health education demand for HPV infection from the perspective of the disease itself, the patient's family system, social system, surrounding environment, psychology and other internal and external systems. After systematic literature analysis and semi-structured interview, the scale formed an item pool containing 45 initial items, and then assessed and supplemented the items through expert correspondence. Through the feedback of patients on the recognisability and operability of the items in the pre-survey, the descriptive definition of the items was improved, so that the items had the best suitability. The experts consulted in this study covered various fields such as cervical diagnosis and treatment, gynecological nursing, nursing management, nursing psychology, nursing education, and were representative to a certain extent. After two rounds of expert consultation and exploratory factor analysis, the scale items were screened from the initial 45 to 28. It basically conforms to the construction principle of the scientific scale "The number of initial items should be at least 50% more than the number of formal items" [32].

In this study, based on the KANO model [15], the positive and reverse transformation of the questions on the scale items was carried out, and the reliability and validity of the positive and reverse items were analyzed respectively. The analysis results showed that the positive and reverse items all had good reliability and validity. Although the reliability of the subscale in the retest reliability did not reach 0.7, it may be related to the small number of subscale items. However, the retest reliability of the positive items total table reached 0.984, and the retest reliability of the reverse items was 0.697. In the exploratory factor analysis, the factor structure of the positive items was roughly the same as that of the reverse items, and the variance contribution rate is both > 60%. After expert discussion, the factor structure is determined as five factors, which basically covers micro and medium-sized systems in ecosystem theory.

The scale of health education demand for patients with HPV infection will have good clinical practicability. Demand-oriented health education is to carry out targeted health education strategies on the basis of the evaluation of the contents, methods and acceptability of patients' health education, and maximize the effect of health education. In recent years, demand-oriented health education has developed rapidly, involving breast cancer patients [33], pregnant woman [34], and other patients. Relevant demand theories and models have also been applied to patients' needs assessment, such as Omaha system theory [35], Maslow's hierarchy of needs theory [33], KANO model [36], The need assessment tools developed on the basis of the need theory are conducive to diversified understanding of patients' needs. This paper provides a

reference for further developing the content and form of health education oriented to the needs of patients.

Health education based on patients' needs can make better use of effective medical resources and precision health education. Most of the known questionnaires on health education needs are self-made by researchers, and there are few systematically developed questionnaires on health education needs, among which the more mature ones are in the field of breast cancer. Developed on the theoretical basis of Maslow's hierarchy of needs theory and the "bio-psycho-social" medical model, there is no mature HPV health education needs questionnaire. The theoretical structure of the health education needs questionnaire for HPV infected patients constructed in this study is based on the ecosystem theory, covering individual, family, environment and other aspects, with a comprehensive assessment range. At the same time, KANO model is combined to analyze the demand categories of patients and determine the priority of demand, which is conducive to better use of medical resources.

## Limitation

The population in this study is only focused on one region, which is limited and the representative sample is insufficient. In addition, the questionnaire collection method in this study is online, which may not represent the real needs of patients due to the deviation of personal understanding of the questionnaire items.

## Conclusion

In this study, a three-stage design was adopted to compile a questionnaire on the health education demand of patients with HPV infection, and based on the KANO model, the items were transformed into positive and negative ones. The research found that the positive and negative questionnaires had good reliability and validity, and could be used as a tool to understand the health education demand of patients with HPV infection. At the same time, the positive and negative questions could clearly analyze the types of patients' needs. It includes attractive demand, must-be demand, one-dimensional demand, Indifferent demand and reverse demand, so that patients can be timely and targeted for health education. In the future, this questionnaire should be applied to patients in more regions to verify its effectiveness.

## Supporting information

**S1 Fig. Standardized pathway coefficient plot after CFA for the final model (positive items).** aa: fl = disease information demand; f2 = social support; f3 = emotional demand; f4 = family support; f5 = health education style demand.
(TIF)

**S1 Data. The data.**
(XLSX)

**S1 File. The health education demand scale for HPV infected patients based on KANO model.**
(DOCX)

## Acknowledgments

We are very grateful to the Patient Follow-up Center of West China Second Hospital of Sichuan University for its contribution to the collection of questionnaires. We would like to thank

Yi Yang, Changsha Yin, Yuqin Tu, Siqi Li for valuable contributions to this research, including collaboration in experiments and data collection.

## Author Contributions

**Data curation:** Yuying Zhang, Die Hou.

**Formal analysis:** Yaling Zhang.

**Investigation:** Yuying Zhang, Li Luo, Shuaihui Du, Die Hou.

**Methodology:** Li Luo.

**Project administration:** Yuying Zhang, Lihua Zhou.

**Resources:** Shuaihui Du.

**Validation:** Yaling Zhang.

**Visualization:** Limin He.

**Writing – original draft:** Yuying Zhang.

**Writing – review & editing:** Lihua Zhou.

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
