## [Decision Letter · Decision Letter 0]

15 Sep 2024

PONE-D-24-32314Development and validation of the health education demand scale for HPV infected patients based on KANO modelPLOS ONE

Dear Dr. Zhou,

Thank you for submitting your manuscript to PLOS ONE. After careful consideration, we feel that it has merit but does not fully meet PLOS ONE’s publication criteria as it currently stands. Therefore, we invite you to submit a revised version of the manuscript that addresses the points raised during the review process.

We look forward to receiving your revised manuscript.

Kind regards,

Nicola Serra

Academic Editor

PLOS ONE

Journal Requirements: When submitting your revision, we need you to address these additional requirements. 1. Please ensure that your manuscript meets PLOS ONE's style requirements, including those for file naming. The PLOS ONE style templates can be found at https://journals.plos.org/plosone/s/file?id=wjVg/PLOSOne_formatting_sample_main_body.pdf and https://journals.plos.org/plosone/s/file?id=ba62/PLOSOne_formatting_sample_title_authors_affiliations.pdf 2. PLOS requires an ORCID iD for the corresponding author in Editorial Manager on papers submitted after December 6th, 2016. Please ensure that you have an ORCID iD and that it is validated in Editorial Manager. To do this, go to ‘Update my Information’ (in the upper left-hand corner of the main menu), and click on the Fetch/Validate link next to the ORCID field. This will take you to the ORCID site and allow you to create a new iD or authenticate a pre-existing iD in Editorial Manager. 3. We notice that your supplementary figures are uploaded with the file type 'Figure'. Please amend the file type to 'Supporting Information'. Please ensure that each Supporting Information file has a legend listed in the manuscript after the references list.

**Additional Editor Comments:**

Dear authors,

some of my suggestions for improving the manuscript are given below.

1. Statistical analysis section.

This section is the key to understanding the statistical analyses performed in the manuscript. Authors should specify how numerical and qualitative data were described. For example, in the case of numerical data, authors should report, the numerical data were reported as mean ad standard deviation, or median and interquartile range [Q1, Q3], in the case of non-normally distributed data. Furthermore, they should report which statistical tests were used and for which analyses. Finally, they should conclude the paragraph by indicating what level of p-value was chosen to consider a test significant, e.g. All tests with p-value <0.05 were considered as significant.

2. A brief description of Tables 3 and 4 in the results should be made

3. Sample and setting section. The authors should better describe how they estimated the sample size, including the values of the parameters used for the calculation.

Finally, we note that some of the comments of the Reviewers refer to specific articles for you to cite. Please note that it is not mandatory that you cite these specific articles, and you are welcome to seek alternative manuscripts in the literature that are relevant to your manuscript’s content

Best regards.

Reviewers' comments:

Reviewer's Responses to Questions

**Comments to the Author**

1. Is the manuscript technically sound, and do the data support the conclusions?

Reviewer #1: Yes

Reviewer #2: Yes

2. Has the statistical analysis been performed appropriately and rigorously? 

Reviewer #1: I Don't Know

Reviewer #2: I Don't Know

3. Have the authors made all data underlying the findings in their manuscript fully available?

Reviewer #1: Yes

Reviewer #2: Yes

4. Is the manuscript presented in an intelligible fashion and written in standard English?

Reviewer #1: Yes

Reviewer #2: Yes

5. Review Comments to the Author

Reviewer #1: GENERAL COMMENT:

The manuscript aimed to develop and validate a health education needs assessment for patients with HPV infection using the KANO model. The final product is a fascinating picture of the types of health education needs and the influencing factor of patients with HPV infection to create personalized health education programs.

I only suggest a few tips to improve the quality of the manuscript, such as revising the English language and emphasizing some topics.

INTRODUCTION:

Human papillpn virus (HPV) is a virus with multiple genotypes, which is closely related to the degree of human reproductive system lesions and the development of malignant tumors.

Please correct the virus name.

It seems that your introduction is too short. Please emphasize more topics, such as the impact of HPV on cervical lesions and the socio-demographic profile of people with sexually transmitted infections. I invite the authors to consider the following works:

Calagna G. 'Secondary prevention' against female HPV infection: literature review of the role of carrageenan. Expert Rev Anti Infect Ther. 2020

Fasciana T. Socio-Demographic Characteristics and Sexual Behavioral Factors of Patients with Sexually Transmitted Infections Attending a Hospital in Southern Italy. Int J Environ Res Public Health. 2021

Reviewer #2: Manuscript ID: PONE-D-24-32314 Date: 14/09/2024

Title of Manuscript: Development and validation of the health education demand scale for HPV infected patients based on KANO model

GENERAL COMMENT:

The manuscript contributes to understanding the HPV health education needs from the perspective of the disease, the patient's family, and social, environmental, psychological and other internal and external systems. I believe that the workflow is well-designed and well-structured, but I would like to raise some issues you can find below.

I suggest English language editing.

INTRODUCTION:

Page 1: “Human papillpn virus”

Please, correct the spelling.

Page 1: “..degree of human reproductive system lesions..”

What do you mean by this expression?

Page 1: “…cervical cancer is also gradually getting younger...”

I understand that you meant that the age of patients suffering from cervical cancer is getting lower, but I believe you should rephrase the sentence. It is not clear enough as it is.

Moreover, please put a reference attesting to this data.

If you believe it is appropriate, you could expand your references list by citing the following work:

Bosco, L., et al. Potential impact of a nonavalent anti HPV vaccine in Italian men with and without clinical manifestations. Sci Rep 11, 4096 (2021). https://doi.org/10.1038/s41598-021-83639-6

6. PLOS authors have the option to publish the peer review history of their article (what does this mean?). If published, this will include your full peer review and any attached files.

Reviewer #1: No

Reviewer #2: No

---

## [Author Response · Author response to Decision Letter 0]

16 Oct 2024

Additional Editor Comments:

1. Statistical analysis section.

This section is the key to understanding the statistical analyses performed in the manuscript. Authors should specify how numerical and qualitative data were described. For example, in the case of numerical data, authors should report, the numerical data were reported as mean ad standard deviation, or median and interquartile range [Q1, Q3], in the case of non-normally distributed data. Furthermore, they should report which statistical tests were used and for which analyses. Finally, they should conclude the paragraph by indicating what level of p-value was chosen to consider a test significant, e.g. All tests with p-value <0.05 were considered as significant.

Responses：

Thanks to the editors for their valuable suggestions. The description of quantitative and qualitative data in statistical analysis has been supplemented in the data analysis section of the paper. Qualitative data is expressed as frequency and component ratio (%), and quantitative data conforming to normal distribution is expressed as mean ± standard deviation. P<0.05 was considered statistically significant.

2.A brief description of Tables 3 and 4 in the results should be made

Responses：

Thanks for the valuable suggestions provided by the editor. We have made a brief description of table 3 and table 4 in the paper. Table 3 is the factor analysis result of the positive items, table 4 is the factor analysis result of the reverse items, and we finally focus on the result of table 3 (factor analysis result of the positive items).

3.Sample and setting section. The authors should better describe how they estimated the sample size, including the values of the parameters used for the calculation.

Responses：

 Thanks for the editor's valuable suggestions. We have explained the sample size in the paper. According to Kendall's scale preparation sample size setting rule, the sample size is set to 5-10 times of the scale items, and the factor analysis sample size is set to 10-20 times of the scale items, and 10% sample shedding rate is also considered.

Reviewers' comments:

2.Has the statistical analysis been performed appropriately and rigorously?

Response :The data was analyzed using IBM SPSS Statistical window (version 20.0), and IBM SPSS AMOS (version 23.0). Our data analysis strictly conforms to the steps of the scale compilation, and the collected data are analyzed by project analysis, retest analysis and reliability and validity test, among which the validity test included content validity and structure validity.

Reviewer #1: GENERAL COMMENT:

The manuscript aimed to develop and validate a health education needs assessment for patients with HPV infection using the KANO model. The final product is a fascinating picture of the types of health education needs and the influencing factor of patients with HPV infection to create personalized health education programs.

I only suggest a few tips to improve the quality of the manuscript, such as revising the English language and emphasizing some topics.

INTRODUCTION:

Human papillpn virus (HPV) is a virus with multiple genotypes, which is closely related to the degree of human reproductive system lesions and the development of malignant tumors.

Please correct the virus name.

It seems that your introduction is too short. Please emphasize more topics, such as the impact of HPV on cervical lesions and the socio-demographic profile of people with sexually transmitted infections. 

Response：

 Thanks for the reviewer's suggestions. We have changed the virus name in the article, and we have added the impact of HPV on cervical lesions and the socio-demographic profile of people with sexually transmitted infections. 

Reviewer #2: GENERAL COMMENT:

The manuscript contributes to understanding the HPV health education needs from the perspective of the disease, the patient's family, and social, environmental, psychological and other internal and external systems. I believe that the workflow is well-designed and well-structured, but I would like to raise some issues you can find below.

I suggest English language editing.

INTRODUCTION:

Page 1: “Human papillpn virus”

Please, correct the spelling.

Response：

Sorry for the spelling error, the spelling of HPV virus has been corrected. Thanks.

Page 1: “..degree of human reproductive system lesions..”

What do you mean by this expression?

Response：

We are very grateful to the reviewer for raising such questions, and we have revised the expression to mean that HPV infection is related to the severity of cervical lesions.

Page 1: “…cervical cancer is also gradually getting younger...”

I understand that you meant that the age of patients suffering from cervical cancer is getting lower, but I believe you should rephrase the sentence. It is not clear enough as it is.

Moreover, please put a reference attesting to this data.

Response：

 Thanks for the reviewer's suggestion, we have modified the expression to make it better understood. In addition, we also supplement the relevant evidence. Thanks.

---

## [Decision Letter · Decision Letter 1]

30 Oct 2024

PONE-D-24-32314R1Development and validation of the health education demand scale for HPV infected patients based on KANO modelPLOS ONE

Dear Dr. Zhou,

Thank you for submitting your manuscript to PLOS ONE. After careful consideration, we feel that it has merit but does not fully meet PLOS ONE’s publication criteria as it currently stands. Therefore, we invite you to submit a revised version of the manuscript that addresses the points raised during the review process.

Dear Authors, the changes you have made have improved the manuscript. There are only a few comments from Reviewer 2 that the authors should address before the manuscript can be accepted for publication. Best regards Nicola Serra==============================

We look forward to receiving your revised manuscript.

Kind regards,

Nicola Serra

Academic Editor

PLOS ONE

Journal Requirements:

Reviewers' comments:

Reviewer's Responses to Questions

**Comments to the Author**

1. If the authors have adequately addressed your comments raised in a previous round of review and you feel that this manuscript is now acceptable for publication, you may indicate that here to bypass the “Comments to the Author” section, enter your conflict of interest statement in the “Confidential to Editor” section, and submit your "Accept" recommendation.

Reviewer #1: All comments have been addressed

Reviewer #2: All comments have been addressed

2. Is the manuscript technically sound, and do the data support the conclusions?

Reviewer #1: Yes

Reviewer #2: Yes

3. Has the statistical analysis been performed appropriately and rigorously? 

Reviewer #1: I Don't Know

Reviewer #2: I Don't Know

4. Have the authors made all data underlying the findings in their manuscript fully available?

Reviewer #1: Yes

Reviewer #2: Yes

5. Is the manuscript presented in an intelligible fashion and written in standard English?

Reviewer #1: Yes

Reviewer #2: Yes

6. Review Comments to the Author

Reviewer #1: I believe that the manuscript is well-designed and well-structured, and the quality has been improved.

Reviewer #2: The authors have accurately followed the suggestions. However, I would like to make some notes.

INTRODUCTION:

Page 1: “The higher infection rate of HPV, the more serious of cervical precancerous lesions.”

Can I suggest rephrasing the sentence? It would be clearer to exchange the order in something like: “More severe precancerous lesions are associated with higher rates of HPV infection”

Page 1: “Additionally, the age-specific distribution of HPV infection showed a bimodal curve, and the trend of younger age was obvious”

I believe that you should better explain this observation. What do you mean specifically with “the trend of younger age was obvious”? It seams quite a vague comment.

7. PLOS authors have the option to publish the peer review history of their article (what does this mean?). If published, this will include your full peer review and any attached files.

Reviewer #1: No

Reviewer #2: No

---

## [Author Response · Author response to Decision Letter 1]

7 Nov 2024

Reviewers' comments:

Reviewer #2: The authors have accurately followed the suggestions. However, I would like to make some notes.

INTRODUCTION:

Page 1: “The higher infection rate of HPV, the more serious of cervical precancerous lesions.”

Can I suggest rephrasing the sentence? It would be clearer to exchange the order in something like: “More severe precancerous lesions are associated with higher rates of HPV infection”

Page 1: “Additionally, the age-specific distribution of HPV infection showed a bimodal curve, and the trend of younger age was obvious”

I believe that you should better explain this observation. What do you mean specifically with “the trend of younger age was obvious”? It seams quite a vague comment.

Response :Thank you for your valuable suggestions. For the expression of “The higher infection rate of HPV, the more serious of cervical precancerous lesions.”, we have changed the expression. For the meaning of “the trend of younger age was obvious”, we also made changes to make it clearer.

---

## [Decision Letter · Decision Letter 2]

3 Dec 2024

Development and validation of the health education demand scale for HPV infected patients based on KANO model

PONE-D-24-32314R2

Dear Dr. Zhou,

We’re pleased to inform you that your manuscript has been judged scientifically suitable for publication and will be formally accepted for publication once it meets all outstanding technical requirements.

Kind regards,

Nicola Serra

Academic Editor

PLOS ONE

Additional Editor Comments (optional):

Dear Authors,

congratulations, your article has been accepted.

Best regards,

Nicola Serra

Reviewers' comments:

Reviewer's Responses to Questions

**Comments to the Author**

1. If the authors have adequately addressed your comments raised in a previous round of review and you feel that this manuscript is now acceptable for publication, you may indicate that here to bypass the “Comments to the Author” section, enter your conflict of interest statement in the “Confidential to Editor” section, and submit your "Accept" recommendation.

Reviewer #2: All comments have been addressed

2. Is the manuscript technically sound, and do the data support the conclusions?

Reviewer #2: Yes

3. Has the statistical analysis been performed appropriately and rigorously? 

Reviewer #2: I Don't Know

4. Have the authors made all data underlying the findings in their manuscript fully available?

Reviewer #2: Yes

5. Is the manuscript presented in an intelligible fashion and written in standard English?

Reviewer #2: Yes

6. Review Comments to the Author

Reviewer #2: The authors have answered all the issues I previously raised, I do not have any other comment to make.

7. PLOS authors have the option to publish the peer review history of their article (what does this mean?). If published, this will include your full peer review and any attached files.

Reviewer #2: No

---

## [Editor Report · Acceptance letter]

20 Dec 2024

PONE-D-24-32314R2 

PLOS ONE

Dear Dr. Zhou, 

I'm pleased to inform you that your manuscript has been deemed suitable for publication in PLOS ONE. Congratulations! Your manuscript is now being handed over to our production team.

Kind regards, 

on behalf of

Dr. Nicola Serra 

Academic Editor

PLOS ONE